# Mood Disorders and Gluten: It’s Not All in Your Mind! A Systematic Review with Meta-Analysis

**DOI:** 10.3390/nu10111708

**Published:** 2018-11-08

**Authors:** Eleanor Busby, Justine Bold, Lindsey Fellows, Kamran Rostami

**Affiliations:** 1The School of Allied Health and Community, University of Worcester, Worcester WR2 6AJ, UK; buse1_17@uni.worc.ac.uk (E.B.); l.fellows@worc.ac.uk (L.F.); 2Department of Gastroenterology, Mid-Central District Health Board, Palmerston North Hospital, Palmerston North 4442, New Zealand; kamran.rostami@midcentraldhb.govt.nz

**Keywords:** gluten-related disorders, gluten-free diet, coeliac disease, non-coeliac gluten sensitivity, irritable bowel syndrome, mood disorders, affective disorders, depression, major depressive disorder, mental health, nutrition

## Abstract

Gluten elimination may represent an effective treatment strategy for mood disorders in individuals with gluten-related disorders. However, the directionality of the relationship remains unclear. We performed a systematic review of prospective studies for effects of gluten on mood symptoms in patients with or without gluten-related disorders. Six electronic databases (CINAHL, PsycINFO, Medline, Web of Science, Scopus and Cochrane Library) were searched, from inception to 8 August 2018, for prospective studies published in English. Meta-analyses with random-effects were performed. Three randomised-controlled trials and 10 longitudinal studies comprising 1139 participants fit the inclusion criteria. A gluten-free diet (GFD) significantly improved pooled depressive symptom scores in GFD-treated patients (Standardised Mean Difference (SMD) −0.37, 95% confidence interval (CI) −0.55 to −0.20; *p* < 0.0001), with no difference in mean scores between patients and healthy controls after one year (SMD 0.01, 95% CI −0.18 to 0.20, *p* = 0.94). There was a tendency towards worsening symptoms for non-coeliac gluten sensitive patients during a blinded gluten challenge vs. placebo (SMD 0.21, 95% CI −0.58 to 0.15; *p* = 0.25). Our review supports the association between mood disorders and gluten intake in susceptible individuals. The effects of a GFD on mood in subjects without gluten-related disorders should be considered in future research.

## 1. Introduction

Mood disorders are a global healthcare burden, with 300 million people now suffering from depression worldwide [1]. In 2015, the World Health Organisation (WHO) estimated that 4.4% of the global population were suffering from clinical depression—a 18.4% increase in prevalence since 2005. On top of this, around 61 million antidepressants are prescribed in a single year in the UK alone [2], while depressive disorders were ranked as the largest contributor to global non-fatal health loss, as well as increased suicide risk [3].

Wheat products are now the main source of carbohydrate in the Western diet and contain high amounts of the protein, gluten. In recent years, reports of gastrointestinal and extra-intestinal symptoms, due to gluten-containing foods have been on the increase [4]. Coeliac disease (CD) is characterised by intestinal mucosal damage due to an immune response to gluten peptides, with clinical improvement after following a gluten-free diet (GFD) [5]. This involves the elimination of gluten-containing foods from the diet, such as wheat, rye and barley products. CD affects about 1% of the UK population [6] and its global prevalence is on the rise [7]. Moreover, around 10% of CD patients are affected by psychiatric disorders [8], with a higher proportion of CD patients exhibiting depression compared to the general population [9]. However, CD can manifest in a variety of ways, with symptomatically atypical and silent patient subgroups [10], and hence is thought to be underdiagnosed [5]. Therefore, it is a possibility that CD may be misdiagnosed, as depression for example, due to a lack of classical symptoms.

A growing body of evidence suggests that mood symptoms are associated with a spectrum of gluten-related disorders [9,11,12]. Reports of health improvements after following a GFD in the absence of CD has led to non-coeliac gluten sensitivity (NCGS) becoming increasingly recognised as its own clinical entity [13], with evidence indicating a higher prevalence than CD [14]. In contrast to CD, specific serological markers for NCGS are lacking; only some patients exhibit increased antibodies to gluten peptides and no mucosal damage is generally observed [15]. Nonetheless, in 1956 it was suggested that gluten may be associated with mood and psychiatric symptoms in a case series of subjects without CD [16]. More recently, mood symptoms are frequently reported as a result of wheat ingestion [17] with ‘low mood’ being a common motivation for gluten avoidance [18] in the absence of both CD and wheat allergy. Furthermore, recent clinical studies have found raised gluten-related antibodies in patients with bipolar, major depressive disorder, and schizophrenia [19,20,21], while episodes of acute mania may be associated with increased serum levels of antibodies against gliadin [22]. Hence, there is mounting evidence for a, potentially bidirectional, relationship between gluten sensitivity and psychiatric disorders.

Numerous theories regarding the aetiology of mood symptoms in those with gluten-related disorders exist. One theory suggests that an immune response to gluten may lead to depressive symptoms [23]. Further evidence suggests social exclusion could lead to depression in CD [6] while another study relates mood symptoms to adjusting to the chronic nature of a physical disease in general [24]. Contrary to this, Roos et al. found no relationship between gastrointestinal symptoms and psychological well-being in CD [25], although antidepressants have interestingly been found to reduce abdominal pain in irritable bowel syndrome (IBS) [26]. Conversely, nutritional deficiencies may be a causative factor for reduced mood; for instance, B-vitamin supplementation was found to significantly improve depression in adults with longstanding CD on a GFD [27]. Finally, the ingestion of FODMAPs (Fermentable Oligo-, Di-, Mono-saccharides And Polyols—short chain carbohydrates also present in wheat, rye and barley, as well as beans, pulses and certain vegetables), have also been suggested to increase both physical and psychological symptoms in those thought to be gluten sensitive [28,29]. Hence, there appears to be a complex and multifactorial relationship between mood and gluten-related disorders.

Regardless, a GFD has been shown to improve mental health in susceptible individuals. Significant improvements in mood disorders and psychological well-being have been recognised in patients with CD [30,31,32], IBS [33] and NCGS [34] following a GFD, although the magnitude of improvement is found to be dependent on good dietary adherence [11,35]. Moreover, anti-gliadin IgG antibodies disappeared in NCGS patients [34] and markers of systemic inflammation were reduced in IBS patients [36], as well as healthy mice [37] following initiation of a GFD. Hence, a GFD may reduce inflammation and improve mood, although a relationship between these outcomes remains theoretical. 

Whilst the GFD for autism spectrum disorders has been well reviewed [38,39], other reviews of psychiatric and mood disorders in relation to gluten have focussed on CD and epidemiological, rather than interventional, evidence [12,40]. Meanwhile, a review on extra-intestinal symptoms in NCGS [41] included only one study considering psychiatric outcomes [42]. However, a search of registered protocols did not reveal that any systematic reviews on gluten and mood are planned or currently in progress. Therefore, we conducted a systematic review of prospective studies with a gluten challenge or GFD intervention on the prevalence and/or severity of mood symptoms in patients with or without gluten-related disorders. Our study was underpinned by the following objectives: To establish whether a relationship exists between mood and gluten; to explore the outcomes of severity mood symptoms and the prevalence of mood disorders; to assess the impact of the level of adherence to a GFD on the severity of mood symptoms; to highlight gaps in the research literature; and to determine implications for practice in terms of implementing a GFD in those with gluten-related and mood disorders.

## 2. Materials and Methods

### 2.1. Eligibility Criteria

The eligibility criteria for inclusion of studies into our systematic review consisted of the following:All studies evaluating the effect of gluten ingestion or elimination on the presence or severity of depressive symptoms and other mood disorders, with any gluten-related intervention for any length of time.As evidence suggests a correlation between mood and level of adherence to a GFD [35,43,44], dietary adherence must be defined using a validated measure and either specify good adherence or report data for adherent and non-adherent participants separately.All prospective intervention studies—randomised, non-randomised, longitudinal—which investigated the change in the severity of mood symptoms as a primary or secondary outcome using a validated questionnaire. We excluded retrospective and cross-sectional studies, as we aimed to investigate the relationship between mood and gluten over a specified amount of time while measuring adherence.Published studies in English.

Further to this, we defined the following eligibility criteria for inclusion of study data into meta-analysis:Data must be reported as means and standard deviations (SDs), or these values must be calculatable or estimable using the available data by methods outlined in the Cochrane Handbook [45].Each study should report a different sample; for different studies utilising the same sample, as part of a follow-up study or ad-hoc analysis for example, only data from the most relevant study or the study reporting the largest sample were included.For comparisons between patients and healthy controls, control and patient groups must be demographically matched by age and gender.

### 2.2. Search Strategy and Selection Criteria

The scientific databases CINAHL, PsycINFO, Medline, Web of Science, Scopus and the Cochrane Library were searched using the strategies outlined in Appendix A. These were designed using keywords, Medical Subject Heading (MeSH) terms and free text words, such as gluten-free diet and depression, combined using Boolean operators. The strategies were piloted for each database and three authors (E.B., J.B. and L.F.) reviewed and amended the search strategy before E.B. commenced the final search. To ensure all relevant studies were captured, two authors (E.B. and J.B.) independently screened and selected the studies. In cases of disagreement, a third author (L.F.) was consulted for the final decision. Reference lists of relevant studies were also searched.

### 2.3. Data Extraction and Quality Assessment

One reviewer (E.B.) extracted the data according to a data extraction form developed to collect information regarding study design, population, intervention, controls and outcomes. The data extraction form included information on authors, country, recruitment methods, number of participants, methods of measuring adherence, level of dietary adherence, commercial funding and/or conflicts of interest, study/intervention duration and analysis strategy (ITT—intention-to-treat; PP—per-protocol). Further data was extracted in order to assess risk of bias (ROB) according to tools developed by the Cochrane Collaboration; Cochrane’s ROB 2.0 (University of Bristol, Bristol, UK) [46] was used for randomised controlled trials (RCTs) and the Risk Of Bias In Non-randomized Studies of Interventions (ROBINS-I) tool (University of Bristol, Bristol, UK) [47] for all types of non-randomised studies. The bias domains included in ROBINS-I overlap with the Cochrane ROB 2.0 tool, but instead of assessing the randomisation process, include the additional domains: Confounding, selection of participants into the study and classification at intervention. Specific criteria for assessing the risk of bias in each of the domains in the context of our review are described in Appendix A.

### 2.4. Statistical Analysis

We performed all meta-analyses with Review Manager (RevMan) 5.3 (The Nordic Cochrane Centre, Copenhagen, Denmark, 2014). Outcomes are based on random-effects models using mean differences. Results from the analysis are presented as mean differences along with the 95% confidence intervals. Statistical significance was set at 0.05 for two-sided *p*-values. Data was synthesised by meta-analysis when there was a consistent comparison in two or more studies measuring depressive outcomes. Where possible, only depressive outcome data from questionnaires not biased towards physical illness, with no questions related to gastrointestinal health and eating habits, were synthesised in meta-analysis; data from studies using biased questionnaires containing these types of questions were reported separately. For continuous data, scores from depression scales were reported as means and the standardised mean difference (SMD) was used as a summary statistic. The means of psychometric scales that increased with severity of depression were multiplied by −1 to ensure that all the scales point in the same direction. Dichotomous data were presented as the percentage of patients who were depressed with a score above a specified cut-point, which we reported only when the cut-point score used in the study was based on a validated rather than arbitrary figure. Risk difference (RD) was used to report the results as this is a measure of absolute effect and more intuitive to interpret [45], especially for change of scores from baseline. Funnel plots were used to evaluate publication bias. All forest plots were stratified according to risk of detection bias. We have highlighted this domain as the key risk of bias domain for our study due to our outcome of interest being a subjective measurement. We used the Grades of Recommendation, Assessment, Development and Evaluation Working Group (GRADE Working Group) system [48,49] for grading the quality of evidence for each outcome according to study design, consistency, directness, imprecision and reporting bias. We used GRADEpro GTD to build the Summary of Findings (SoF) and GRADE profile tables [50].

### 2.5. Heterogeneity and Sensitivity Analyses

Heterogeneity between studies was interpreted according to general guidance [51], which suggest that *I*^2^ values of 25%, 50%, and 75% may indicate low, medium and high heterogeneity, respectively, while a value of 0% indicates no observed heterogeneity. To address the most important sources of heterogeneity, we performed planned subgroup analyses considering the effect of CD (CD vs. non-CD participants), gastrointestinal symptoms (symptomatic vs. asymptomatic), and the presence of the CD-associated HLA-DQ2/8 genotype (positive vs. negative) on depressive outcome measures. We also retrospectively performed a subgroup analysis into population sample country of origin after extracting and analysing the data in order to further investigate heterogeneity. The effect of dietary adherence (compliant vs. noncompliant participants) was analysed as a separate comparison so as to include data from Nachman et al. (2010). Sensitivity analyses were performed for: Study searching, by including abstracts whose results could not be confirmed in subsequent publications; data selection, by excluding results from studies utilising an ITT approach; study methods, by analysing only studies with an average intervention time of one year; study quality, by excluding studies at a high risk of bias in key domain(s); and analysis methods by changing random-effects (RE) for fixed-effects (FE) and risk differences (RD) for risk ratios (RR) and odds ratios (OR).

### 2.6. Missing Data

All studies reporting the necessary outcome data as mean values with SDs were included in meta-analysis. If the necessary data was not reported in the correct format for meta-analysis, the corresponding author of the relevant study was contacted via email to request the required data. If no reply was received, a reminder email was sent three weeks after the initial request and other study authors were contacted if emails could be retrieved. As a final resort, and if possible, means and SDs were calculated from the available information (as long as the data were determined to be normally distributed) according to the methods outlined in Chapter 7 of the Cochrane Handbook version 5.1.0 [45]. Some scores were derived from graphs by optical plot reading using WebPlotDigitizer (Version 4.1, Ankit Rohatgi, Austin, TX, USA) [52]. 

## 3. Results

### 3.1. Literature Search

The final literature search for all databases was conducted on 8 August 2018 by E.B. These searches identified 236 papers, three additional citations were identified by a recursive bibliography search [34,53,54], one study was referred by an expert in the field [55] and one study was already known by the authors [29]. After excluding records based on duplicate data, title or abstract, fifty-one were fully reviewed. Finally, 13 studies met the inclusion criteria (Figure 1). The characteristics of these included studies are summarised in Table 1. A summary of reasons for the studies excluded by full-text screening are provided in Appendix A.

### 3.2. Characteristics of the Included Studies

Of the 13 studies, three were RCTs [42,53,56]. Study participants were subjects with self-reported NCGS with [42] and without [56] diagnosed IBS, and asymptomatic EmA-positive subjects [53]. Each of these studies excluded CD either by previous diagnosis [53] or study screening [42,56]. The remaining 10 studies were single arm before-after studies; one of these was a time-interrupted study and one was a follow-up study. Of these, two reported on the same group of participants; Nachman et al. (2009) reported the initial study period from baseline to one year at intervals of three months for all participants, while Nachman et al. (2010) reported baseline, one-year and four-year follow-up results for a subset of the same participants (*n* = 53) who continued to follow a GFD for the full four years. Although using the same sample, both studies were included as they elucidate the short and long-term effects of a GFD on depressive symptoms. Nevertheless, results from these studies were not pooled in meta-analysis as they report on the same participants, and hence were analysed separately.

There were differences in the questionnaires utilised to measure the severity of depression between the included studies. Three studies used questionnaires containing questions related to gastrointestinal health and eating habits; two used the Beck Depression Inventory (BDI) [57,58], and one the Hamilton Depression Rating Scale (HDRS) [59]. However, one of these studies reported some data from a sub-analysis removing these questions [58]. Conversely, nine studies used questionnaires containing no such questions; four used the Psychological General Well-Being Index (PGWB) [53,60,61,62], one the Hospital Anxiety and Depression Scale (HADS) [63], one the Crown-Crisp Experiential Index (CCEI) [64], one the Children’s Depression Inventory (CDI) [11] and one the State-Trait Personality Inventory (STPI) [42], while one further study [65] used a modified version of the Zung Self-Rating Depression Scale (SDS) with such questions removed. Finally, one study simply asked participants to grade depression as present or absent on each day [56]. 

### 3.3. Quality Assessment

Of the RCTs, one study was found to have a low risk of bias and two were found to have a high risk of bias. Of the non-randomised studies, three were found to have a moderate risk of bias while the remaining seven were found to have a serious risk of bias. Graphical representations of the summary of risk of bias for individual studies and across all studies are given in Figure 2, and the analysis of each domain is detailed in Appendix A.

### 3.4. Data and Analyses

A summary of our findings are presented in Appendix A.

#### 3.4.1. A GFD vs. A Gluten-Containing Diet

##### 3.4.1.1. Difference in Mean Depression Scores

One RCT compared a GFD to a normal gluten-containing diet in a two-arm study [53] and ten reports of nine single-arm before-after studies compared depression scores for participants at baseline (on a gluten-containing diet) and after following a GFD. However, two studies were not eligible for meta-analysis [59,65] and one study [57] was a follow-up of the same sample from another study [58]. Hence, eight studies with a total of 953 participants were included in meta-analysis. We found that a GFD significantly reduced depressive symptoms in 953 participants overall (SMD −0.37, 95% CI −0.55 to −0.20; *p* < 0.0001) (Figure 3A). There was low-medium heterogeneity between the studies overall (*I*^2^ = 38%), but zero heterogeneity between the non-randomised studies and RCTs. Subgroup analyses (Appendix A) revealed no difference in effect between those with and without diagnosed CD (*p* = 0.73) or between those HLA-DQ2/8-positive and HLA-DQ2/8-negative (*p* = 0.49). Conversely, there was a significant difference in effect between those with classical, atypical and silent CD (*p* = 0.003) with high heterogeneity between subgroups (*I*^2^ = 82.5%) (Figure 3B); while classical CD patients exhibited a significant effect (SMD −0.65, 95% CI −0.96 to −0.34; *p* < 0.0001), the effect for silent CD patients was nonsignificant (SMD −0.06, 95% CI −0.38 to 0.26; *p* = 0.71) and one study reported a significant effect for atypical CD patients. 

Of the studies not eligible for meta-analysis, one reported non-normally distributed data as medians and IQRs as opposed to means and SDs [59] and one only reported mean scores for patients positive for depression rather than all participants [65]. One study included in meta-analysis did not report SDs for the PGWB subcategories [61], so SDs for the mean depression values were estimated using methods described in the Cochrane Handbook [45] and 95% CI values imputed from an ad-hoc analysis of the same population sample [66]. Only data for the classical CD patients from Nachman et al. [58] were used because this was the only data reported for the modified BDI removing questions to avoid bias due to illness. Moreover, only scores for adherent participants (*n* = 7) were used from Simsek et al. [11] as noncompliance was high with 17/24 participants (71%) found not to follow a strict GFD. Finally, one study providing data for a second follow-up at four years [57] suggests an insignificant trend towards worsening depressive symptoms relative to one year (*p* = 0.39), which remained significant relative to baseline (*p* < 0.0001) (Appendix A).

##### 3.4.1.2. Change in Number of Participants Positive for Depression

Four reports of three studies compared the number of participants positive for depression at baseline and after following a GFD. Three were included in meta-analysis (Figure 4). For 110 classical CD patients, there was a reduction of 31% of patients positive for depression after following a GFD (RD −0.31, 95% CI −0.52 to −0.10; *p* = 0.003). All the included data is for classical CD patients following a GFD for one year; no studies reported this outcome for non-CD patients. We found a significant difference in effect between the studies reporting results from modified and unmodified questionnaires, with a significant effect seen for the unmodified questionnaires and a nonsignificant effect in studies using an unmodified questionnaire.

#### 3.4.2. Gluten Challenge vs. Placebo (GFD)

Two RCTs compared the mean depression scores of subjects during the gluten and placebo challenge periods and were included in meta-analysis [42,56] (Figure 5). We found a trend towards worsened depressive symptoms during the gluten challenge period compared to placebo, although this did not reach significance (SMD 0.21, 95% CI −0.58 to 0.15; *p* = 0.25). Heterogeneity was low (*I*^2^ = 19%). Both used a cross-over trial design with participants acting as their own controls and both adequately described blinding of participants and researchers. Moreover, both used a per-protocol (PP) approach in their analyses. Depression scores were derived from a graph by optical plot reading using WebPlotDigitizer [52]. 

#### 3.4.3. Compliant vs. Noncompliant Participants

Three publications of two studies reported depression scores separately for CD patients compliant and noncompliant to the GFD; no studies reported separate data for these subgroups in non-CD patients. However, two of the three studies used the same sample at different follow-up timepoints [57,58]. Moreover, as there is considerable variation in results and inconsistency in the direction of effect, no meta-analysis was conducted (Figure 6). Firstly, Nachman et al. [58] found a nonsignificant difference in mean depression scores between 59 compliant (M −7.9, 95% CI 4.8 to 11.0) and 25 noncompliant patients (M −6.3, 95% CI 3.6 to 9.5) at year one, although there was a slightly lower severity of depressive symptoms in the noncompliant subgroup. Nevertheless, they reported a significantly higher severity of depression in the noncompliant subjects after four years on a GFD (*p* = 0.04), with 27 compliant and 26 noncompliant CD patients having mean scores of 5.8 (95% CI 2.1 to 9.5) and 11.3 (95% CI 7.6 to 15.0), respectively [57]. Conversely, when comparing the number of patients positive for depression (Appendix A), they consistently reported a nonsignificant trend towards a lower number of depressed patients in the compliant group, with no difference between the proportion of depressed patients at one year and four year follow-ups (*p* = 0.86; *I*^2^ = 0%). On the contrary, Simsek et al. [11] found the severity of depressive symptoms to be significantly higher in noncompliant, compared to compliant, CD children after only one year on a GFD (*p* = 0.005), with seven compliant patients and 17 noncompliant patients achieving mean CDI scores of 4.75 (SD 3.3) and 12.33 (SD 5.8), respectively.

#### 3.4.4. GFD-Treated Patients vs. Healthy Controls

##### 3.4.4.1. Difference in Mean Depression Scores

Five publications reporting on four studies included an eligible healthy control group, of which four were included in meta-analysis (Figure 7A). As Nachman et al. [57,58] used the same sample and healthy controls, the results for the four year follow-up [57] were reported separately. Overall, we found no difference between the depressive outcome scores between 868 GFD-treated patients and 400 healthy controls at one year follow-up (SMD 0.01, 95% CI −0.18 to 0.20, *p* = 0.94) and zero heterogeneity between the studies (*I*^2^ = 0%). Similarly, there was no significant difference at the four year follow-up between 27 strictly adherent patients and 70 healthy controls (SMD −0.08, 95% CI −0.52 to 0.36, *p* = 0.72). Two studies with a healthy control group were not eligible for meta-analysis; Simsek et al. [11] reported depressive outcome data for their healthy controls as medians and IQRs, but similarly reported an insignificant difference between patients and controls, while Collin et al. [64] did not demographically match patients and healthy controls, hence we have not reported their data.

##### 3.4.4.2. Difference in Number Participants Positive for Depression

Three studies [57,58,65] reported the number of GFD-treated participants and healthy controls positive for depression, of which two were meta-analysed and data from Nachman et al. [57] were reported as separately (Figure 7B). Only data from patients strictly adherent to the GFD were included. We found a trend towards an increased prevalence of depression in GFD-treated patients compared to healthy controls at one year (RD 0.21, 95% CI −0.16 to 0.58; *p* = 0.26) and four years (RD 0.10, 95% CI −0.02 to 0.22; *p* = 0.12), although these were nonsignificant. There was no significant difference between the results at one year and four years (*p* = 0.56), and heterogeneity was zero between all studies and subgroups.

### 3.5. Sensitivity Analyses

Various sensitivity analyses were untaken to ensure significant differences were not as a result of arbitrary decisions throughout the study process (Appendix A and Appendix A). Firstly, no significant differences in meta-analysis results were found when interchanging random-effects for fixed-effects, or risk difference for odds ratio and risk ratio, for the majority of analyses (Appendix A). However, there was a difference in the final results for analysis Section 3.4.4.2. at one year, which became significant when using fixed-effects, as opposed to random-effects, and odds ratio or risk ratio, as opposed to risk difference. Secondly, removing studies at an unclear/high risk of detection bias, leaving only those at low risk, did not produce substantially different results for any applicable comparison (Appendix A). Thirdly, while removing data from Nachman et al. [58] from our meta-analyses did not significantly alter the results, overall heterogeneity was reduced from *I*^2^ = 38% to *I*^2^ = 0% for the main analysis (Appendix A) and from *I*^2^ = 52% to *I*^2^ = 13% for the CD subgroup. On the other hand, there was no difference in results or heterogeneity between using outcome data from Nachman et al. for classical CD patients from the modified BDI and the unmodified BDI [58], or all CD patient subgroups from the unmodified BDI [57] (Appendix A). Finally, the results from a potential conference abstract for a cross-over RCT [67], excluded from our study due to the lack of a published final article, could not be included in a sensitivity analysis as only the mean change in the STPI state depression sub-score was reported between gluten and placebo groups (mean change 0.69, 95% CI −2.15 to 3.53, *p* = 0.61). Nevertheless, these results for NCGS patients are in agreement with our meta-analysis in Section 3.4.2.

### 3.6. Publication Bias

Inspection from the funnel plot that arose from our main analysis (Figure 8) suggests the presence of publication bias due to location biases [68], with published studies from Finland less likely to find to find a large effect from a GFD on reducing depressive symptom scores relative to published studies from other countries.

## 4. Discussion

Our systematic review involved a total of 13 studies and 1139 patients, with meta-analysis on an eligible sample of 933 patients from non-randomised studies and 99 patients from RCTs, as well as 180 healthy controls. Although we generally found a low level of heterogeneity, a limitation of our review was the small number of studies available for subgroup analyses that limited our ability to investigate any heterogeneity. Moreover, despite the fact that we contacted authors for missing data, no additional data was retrieved. This was either because: The data was no longer available; authors had retired or moved to another area of research; or a lack of response. Nevertheless, we adhered to the Preferred Reporting Items for Systematic Reviews and Meta-Analyses (PRISMA) statement [69] guidelines wherever possible (see PRISMA checklist, Appendix A) and assessed the quality of the individual studies using tools developed and recommended by the Cochrane Collaboration for both the RCTs and non-randomised studies. Moreover, we applied the GRADE process [48] to assess the certainty of our conclusions and recommendations based on the evidence across the studies for each outcome.

A further strength of our systematic review was our comprehensive search strategy, which we piloted and tailored to numerous databases, and strict application of inclusion and exclusion criteria. Therefore, we are relatively certain that all relevant studies have been included in our review. Although diagnosed conditions or disorders were not an exclusion criteria for our study, only studies on populations with CD, IBS or NCGS were identified through our searches; no other gluten-related disorder, such as dermatitis herpetiformis or gluten ataxia, nor any other condition, such as major depressive disorder, were identified. Moreover, our searches only identified studies assessing depression, or depression as a subcategory of quality of life; no studies were found that assessed other determinants of mood or mood disorders. We conducted further, broader searches for other mood disorders as a sensitivity analysis in attempts to find studies we may have missed in our search strategy, but identified no further relevant studies. As we found no studies that attempted a GFD intervention on a sample of patients with depression, despite evidence for significantly higher levels of gluten-related antibodies in patients with major depressive disorder [19], this would be an interesting topic for future studies to address in order to help assess the directionality of the relationship between depression and gluten.

Our first objective was to establish whether a relationship exists between mood and gluten in those with and without gluten-related disorders. We found that a long-term GFD may significantly reduce and normalise the severity of depressive symptoms for subjects with CD, IBS and NCGS, with a medium-large effect for both symptomatic and atypical CD patients, but no effect for asymptomatic/silent cases [70]. However, the criteria for what constitutes silent CD remains uncertain; although neuropsychiatric disorders are likely to be included in the definition of atypical CD [71], there are a variety of pathophysiological differences underlying the clinical spectrum of depressive disorders [72]. Hence, it is uncertain whether those with depression, but no other symptoms, at baseline would be classified as having atypical or silent CD. Moreover, one of our included studies found that the significant improvement in depressive symptoms for the atypical/silent combined subgroup was no longer significant when the questionnaire was modified to remove questions based on gastrointestinal symptoms [58]. Conversely, another one of our included studies reported that all asymptomatic CD participants randomised to the GFD group for the first year of the study refused to crossover to a follow a gluten-containing diet again, due to a fear of worsening symptoms [53], suggesting that even subjects who did not report any symptoms at baseline experienced improvements after following a GFD. Hence, although we established an overall effect, is it difficult to draw many conclusions based on symptom classification at this time.

Further to this, we assessed the impact of the level of adherence to a GFD on mood symptom severity. Interestingly, we found a significant difference in mean depression scores in favour of strict compliance for CD children after one year, whereas the difference for CD adults was nonsignificant at the same timepoint. This, nevertheless, became significant in favour of compliance at the four year follow-up. Previous systematic reviews and meta-analyses have consistently found a moderate association between poorer GFD adherence and worsened depressive symptoms [35,73], though with high heterogeneity between the studies. However, our nonsignificant finding for adults after the first year does not support this relationship described by others. On one hand, a standardised method for measuring adherence to the GFD does not yet exist, and hence there were differences in the methods utilised by each studies. Alternatively, recent cross-sectional studies suggest that hypervigilance to a GFD, associated with greater knowledge, was significantly associated with reduced quality of life [44], and that those with worse economic status were at an increased risk of lower quality of life while following a GFD [74]. Conversely, the presence of depression has been suggested to weaken the correlation between GFD adherence and symptoms [43], implying that symptoms may be driven by factors other than gluten exposure. To summarise, further studies with standardised measurements of adherence are required before definitive conclusions may be drawn on the effects of gluten-free dietary adherence on the severity of depressive symptoms.

On the other hand, we found the proportion of participants testing positive for depression tended to be higher in GFD-treated patients compared to healthy controls at both one and four years, which was unaffected by the level of compliance. In line with this, previous studies suggest that up to 30% of CD patients show persistent enteropathy after one year on a GFD [75], potentially due to consuming trace amounts of gluten via cross-contamination [76,77]. Despite this, recent RCTs suggest that a low-FODMAPs diet can further reduce the severity of depressive symptoms in those with NCGS [29] and CD [78] already on a GFD, although further research is needed in this area. In addition, while it has been suggested that altered gut microbiota may contribute to the psychiatric effects of a GFD [29,79,80], results should not be extrapolated from one population to another, due to the highly individualised pattern of gut microbial composition [81]. In any case, future studies should be mindful of the shortcomings of only considering mean scores of the sample as a whole, and closer attention should be paid to patients who may be unresponsive to a GFD in research and practice.

In terms of the short-term effects for gluten on mood, the trend towards increased severity of depressive symptoms in NCGS patients after only a few days of a blinded gluten challenge further reinforces our findings that the ingestion of gluten plays a role in the presence of depressive symptoms—even in those without mucosal gut damage. Although one of our included studies reported no concurrent differences in gastrointestinal symptom severity between gluten, whey and placebo challenges [42], other clinical trials on non-CD participants report a significant increase in physical symptoms when challenging with foods containing wheat [82] and fructans [83]. Additionally, despite the fact that another RCT found no gluten-specific gastrointestinal symptoms when challenging NCGS patients already on a low-FODMAPs diet, all patients returned to a GFD at the end of the trial as they subjectively reported “feeling better” [28]. However, the weaknesses of this study have been discussed in a previous paper [84]; while the sample is unlikely to be representative of the NCGS population, the crossover design could have also produced an anticipatory nocebo response [85]. Nonetheless, a proposed mechanism requiring further investigation is that FODMAPs predominantly trigger GI symptoms whereas gluten is a trigger for neurological and psychiatric symptoms by having direct effects on the brain [86].

Unfortunately, the overall quality of the evidence base was poor and confounding factors were problematic. Firstly, while a few studies stated subject antidepressant use as an exclusion criteria, other studies did not consider this. Secondly, seasonal affective disorder (SAD), a type of depression with a seasonal pattern, may overlap with other depressive disorders [87], but was not considered or controlled for in any of the studies. Although the majority of the non-randomised studies planned the follow-up to be one year after the start of the GFD, this timespan varied between clients, as well as between studies and none specified the time of year. Moreover, some of the questionnaires utilised, namely the SDS, HDRS and BDI, contained questions related to gastro-intestinal symptoms and eating habits, which are likely to introduce bias due to physical illness. Finally, our sample is dominated by Finnish participants (75.8%), with only 9.4% participants from Italy and 3.6% from the UK, significantly reducing the applicability of our findings; a GFD may be easier to follow in Finland as there is good knowledge of CD and easy availability of gluten-free products [88], which may lead to a lower risk of depression, due to isolation and other issues associated with following a GFD.

Nonetheless, we set out to determine implications for future research, as well as implementation of GFDs in practice. Firstly, our included studies varied in their criteria for CD diagnosis; whereas one study relied on EmA-positivity [53], another used biopsy [61] as their criteria for inclusion of silent CD patients. Moreover, broad subcategories, such as ‘atypical’, were problematic when attempting to assess specific atypical symptoms, such as depression. Hence, specific standardised criteria for the classification of the different subtypes of CD, as well as other gluten-sensitive disorders, should be developed to aid further research in terms of comparability, as well as identification and suitable treatment of those with CD. Secondly, our finding that the proportion of adults strictly adherent to the GFD decreased significantly over time is supported by a large recent meta-analysis [35], and is likely related to the amount of support received by patients. For instance, a RCT found that six months of psychological counselling improved GFD adherence and reduced depression in CD patients with depression at baseline [89]. While no studies exist that support or repute our findings that a lower proportion of children achieved strict adherence than adults, practical tools have been shown to promote self-management, dietary adherence and well-being in children and adolescents with CD [90]. Hence, the development of both standardised measurement methods and tools to promote dietary adherence would be useful for future research, as well as patient management. Moreover, a balance between dietary adherence and well-being appears important for those following a GFD, with careful consideration of the level of support available for specific populations in maintaining a GFD diet over time.

## 5. Conclusions

Our study confirms that gluten elimination may represent an effective treatment strategy for mood disorders in individuals with gluten-related disorders, while highlighting specific considerations for future research and implications for practice. Firstly, standardisation of methods to measure dietary adherence and mood symptoms with no bias, due to physical illness would greatly increase the validity and comparability of future research. Secondly, future studies focusing on gluten and mood in participants without a gut-related disorder, for example, in a population sample with depression, would contribute to the evidence necessary to determine the directionality of the relationship. Nevertheless, authors should be mindful of the shortcomings of only considering mean scores of a sample, with potentially GFD-resistant participants requiring closer attention. In practice, implementation of a gluten-contamination elimination diet, such as that detailed by Hollon et al. [76], in which processed foods are eliminated, could prove beneficial for some individuals. Thirdly, standardisation of the classification for the subtypes of CD, as well as other gluten-sensitive disorders, should be developed to aid further research in terms of comparability, as well as identification and suitable treatment of those with CD. Finally, the level of support available to help a patient in maintaining a GFD diet over time should be carefully considered when recommending a GFD in practice.

## Figures and Tables

**Figure 1 nutrients-10-01708-f001:**
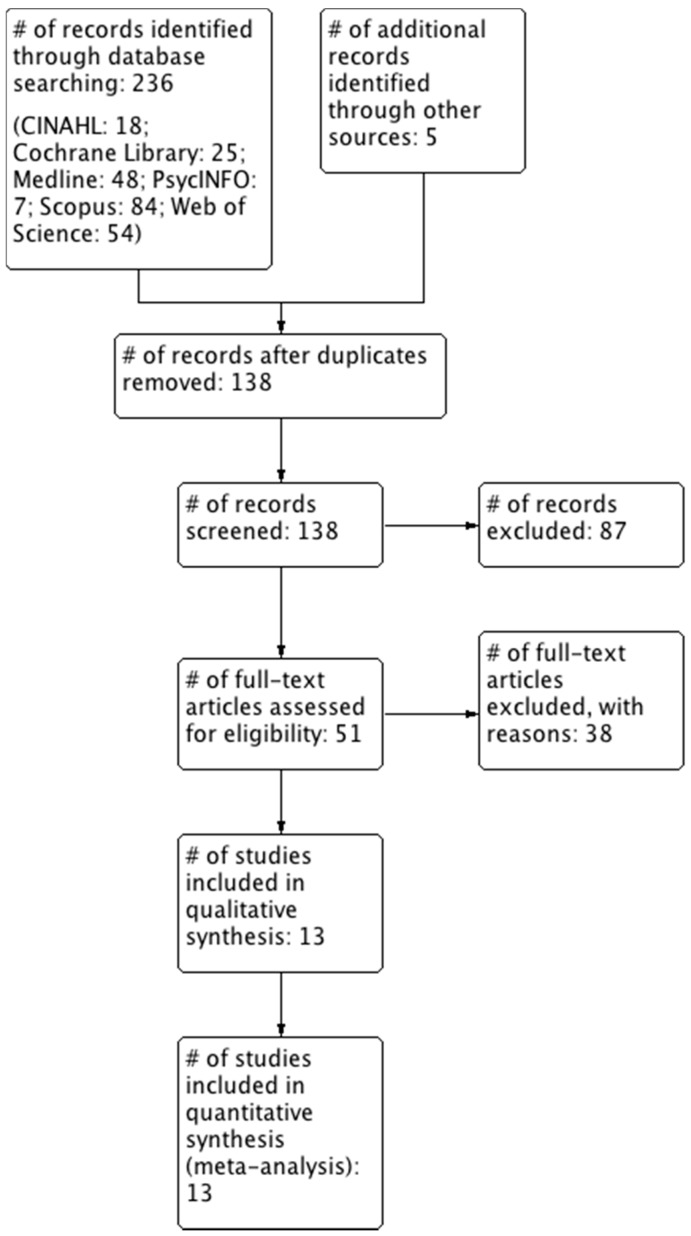
Flow diagram of study selection. #, number.

**Figure 2 nutrients-10-01708-f002:**
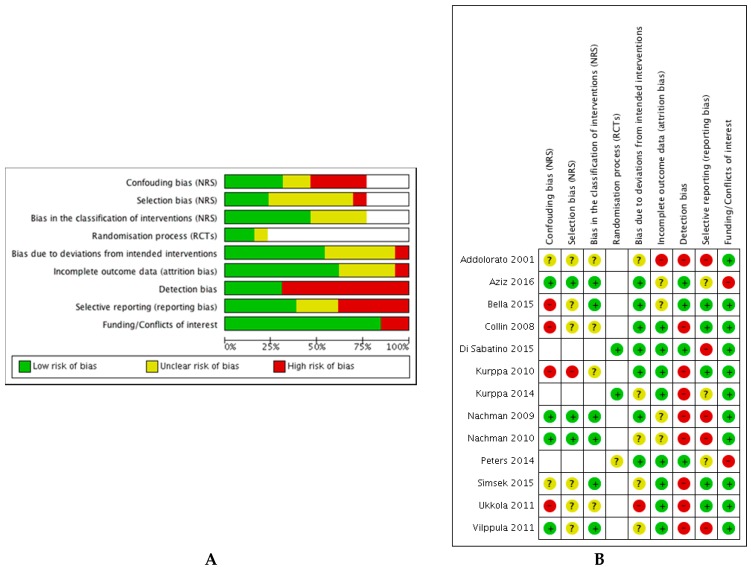
(**A**) Risk of bias graph: Review authors’ judgements about each risk of bias item presented as percentages across all included studies. (**B**) Risk of bias summary: Review authors’ judgements about each risk of bias item for each included study.

**Figure 3 nutrients-10-01708-f003:**
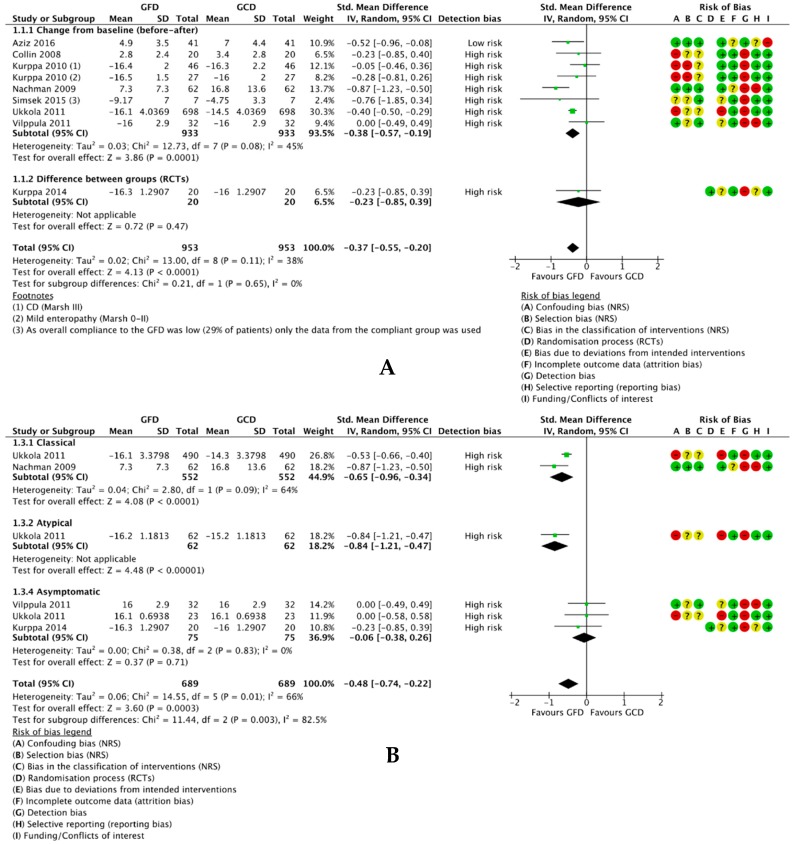
Forest plot demonstrating the difference in mean depression scores between following a GFD and a gluten-containing diet for (**A**) all studies (1 RCT comparing two intervention groups and seven BA studies comparing participant scores at baseline) and (**B**) subgroup analysis based on symptoms (classical, atypical and asymptomatic). CI, confidence interval; df, degrees of freedom; GFD, gluten-free diet; GCD, gluten-containing diet; *I*^2^, heterogeneity; IV, inversed variance; Random, random effects model; SD, standard deviation; Std., standardised; total, number of patients.

**Figure 4 nutrients-10-01708-f004:**
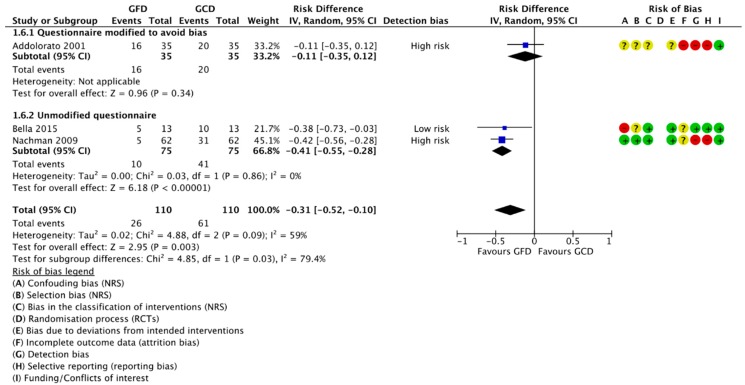
Forest plot demonstrating the change the number of CD participants with depression after following a GFD for one year.

**Figure 5 nutrients-10-01708-f005:**
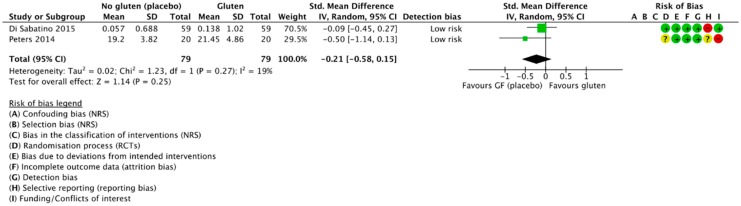
Forest plot demonstrating the difference in participant depression scores between gluten and placebo challenges in subjects with self-reported NCGS.

**Figure 6 nutrients-10-01708-f006:**
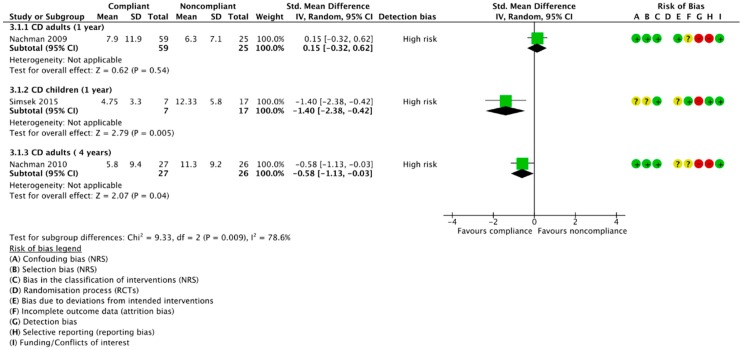
Mean depression scores in compliant vs. noncompliant CD adults at one year, CD children at one year, and CD adults at four years after following a GFD.

**Figure 7 nutrients-10-01708-f007:**
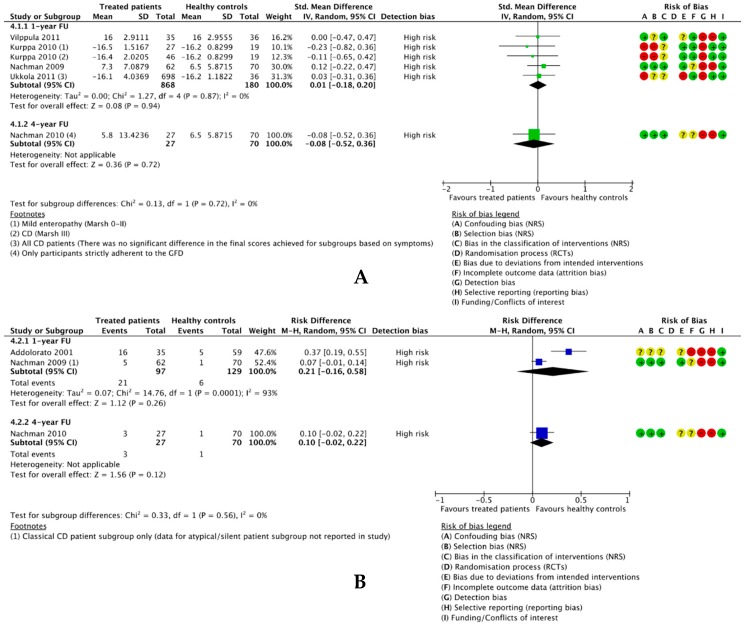
Forest plots comparing: (**A**) Mean depression scores in treated patients and healthy controls and (**B**) the difference in the no. treated patients and healthy controls positive for depression at one year and four years after following a GFD.

**Figure 8 nutrients-10-01708-f008:**
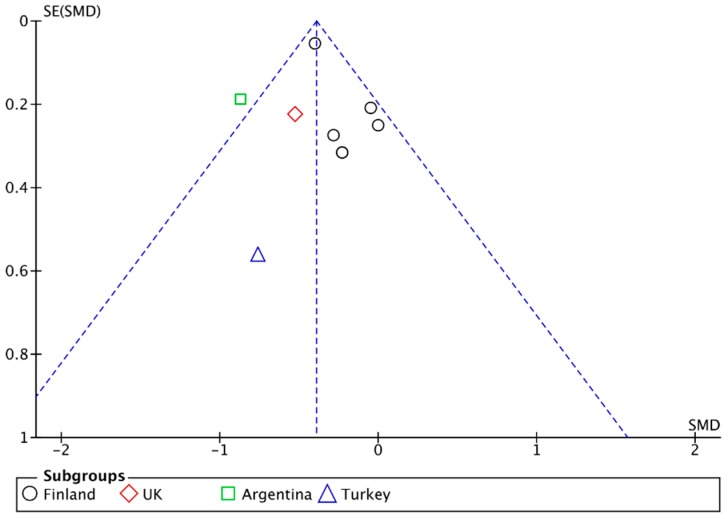
Funnel plot.

**Table 1 nutrients-10-01708-t001:** Characteristics of included studies.

Author (Year)	Country of Origin/Study Design	Participants	Healthy Controls	Interventions	Outcomes	Method of Measuring Adherence *	Notes
Addolorato et al. (2001) [65]	Italy Single centre BA comparison	35 newly diagnosed classical CD patients	59 HC recruited from hospital staff members	GFD for 1 y	Changes in depression score BA a GFD (M-SDS) Changes in no. patients positive for depression BA a GFD (M-SDS score > 49)	1—validated 2—family member interview 3—AGA, EmA 4—biopsy at 6–8 mo	
Aziz et al. (2016) [63]	UK Single centre BA comparison	41 IBS-D patients; 20 HLA-DQ2/8+ and 21 HLA-DQ2/8−	NC	GFD for 6 wk; FU at 18 mo for those who continued on GFD	Changes in HADS before and after a GFD Difference between HLA-DQ2/8+/− groups	1—validated (“Patients scoring 0 or 1 do not follow a strict GFD. Patients scoring 2 follow a GFD but with errors necessitating correction. Finally, patients scoring three or four follow a strict GFD.”)	Educational grant from Dr. Schär (a gluten-free food manufacturer) to undertake investigator-conceived and -led research on gluten sensitivity
Bella et al. (2015) [59]	Italy Single centre BA comparison (FU study from Pennisi et al. (2014))	13 CD patients	NC	GFD for 16 mo	Changes in depression score BA a GFD (HDRS) Changes in no. patients positive for depression (dysthymia) BA a GFD (SCID-I)	1—validated 3—EmA, tTG	
Collin et al. (2008) [64]	Finland Single centre BA comparison	20 biopsy-proven CD patients	HCs not recruited from same community (female students/male workers)	GFD for 1 y	Changes in depression score BA a GFD (sub-score of CCEI/Middlesex Hospital Questionnaire)	3—EmA 4—Vh/CrD	
Di Sabatino et al. (2015) [56]	Italy Multicentre DB, PC, CO RCT	59 patients suspected of having NCGS (CD and WA excluded)	NA	2 arms; GFD (1 wk baseline period) followed by 1 wk: (a) 4.375 g/day gluten (b) rice starch placebo with 1 wk wash-out periods inbetween	Difference in mean daily depression scores (unvalidated questionnaire) between gluten and placebo groups	1—validated 6—unused capsules counted	
Kurppa et al. (2010) [60]	Finland Single centre Cohort	73 EmA-positive adults; 27 mild enteropathy (Marsh I-II), 46 CD (Marsh III)	110 HCs (age and gender matched)	GFD for 1 y	Changes in depression score BA a GFD (sub-score of PGWB)	3—EmA, tTG 4—Vh/CrD	
Kurppa et al. (2014) [53]	Finland Multicentre DB, CO RCT	40 asymptomatic EmA-positive adults (CD excluded)	NA	2 arms; participants randomised to a GFD or gluten-containing diet for 1 y then cross-over	Difference in mean depression score between gluten and GFD groups at 1 y (sub-score of PGWB)	2—NI on who conducted 3—EmA, tTG 4—Vh/CrD	Deviations from protocol: No-one in GFD group willing to restart gluten-containing diet so only data from 1 y FU used
Nachman et al. (2009) [58]	Argentina Single centre BA comparison/time-interrupted study	84 newly diagnosed biopsy-proven CD patients; 62 classical, 14 atypical and 8 asymptomatic	70 HCs recruited from hospital staff members (age and gender matched)	GFD for 1 y	At baseline, 3, 6, 9 and 12 mo: Changes in depression score BA a GFD (BDI) Changes in no. patients positive for depression BA a GFD (BDI score ≥ 18) Subgroup analysis: Compliant vs. noncompliant	Opinion of physician in charge, based on: 1—self-rated questionnaire 2—“meticulous enquiry by an experienced dietician” 3—tTG, DGP, EmA, AGA 4—Vh/CrD 5—4 day (self-reported)	
Nachman et al. (2010) [57]	Argentina Single centre BA comparison/FU from Nachman et al. (2009)	53 CD patients; 37 classical and 16 atypical/asymptomatic	70 HCs recruited from hospital staff members (age and gender matched) (same as Nachman et al. (2009))	GFD for 4 y	At baseline, 1 y and 4 y: Changes in depression score BA a GFD (BDI) Changes in no. patients positive for depression BA a GFD (BDI score ≥ 18) Subgroup analysis: Compliant vs. noncompliant	Opinion of physician in charge, based on: 1—self-rated questionnaire 2—“meticulous enquiry by an experienced dietician” 3—tTG, DGP, EmA, AGA 4—Vh/CrD 5—4 day (self-reported)	
Peters et al. (2014) [42]	Australia Single centre DB, PC, CO RCT	20 IBS patients (CD excluded by biospy); recruited from a preceding study in which subjects with self-reported NCGS were challenged with diets containing varying amounts of gluten (Biesiekierski et al., 2013)	NA	3 arms; low-FODMAPs + GFD (3 day baseline period) followed by 3 day: (a) gluten (16 g/day) (b) whey (16 g/day) (protein control) (c) placebo with 3–14 day washout period in between	Difference in depression scores following each dietary challenge (sub-score of STPI)	1—validated 5—3 day (self-reported) 6—unused/additional food counted	Peter R. Gibson has published two books on a diet for IBS. This study was supported by George Weston Foods as part of a partnership in an Australian Research Council Linkage Project and the National Health and Medical Research Council (NHMRC) of Australia.
Simsek et al. (2015) [11]	Turkey Single centre BA comparison	24 newly diagnosed biopsy-proven paediatric CD patients; age limit 9–16 y	25 HCs recruited from same centre; EmA-negative	GFD for 6–20 mo	Changes in depression score BA a GFD (CDI) Subgroup analysis: Compliant vs. noncompliant	3—EmA, tTG	
Ukkola et al. (2011) [61]	Finland Nationwide BA comparison	698 newly diagnosed (within 1 y) biopsy-proven CD patients; 490 classical, 62 atypical and 146 screen-detected	110 HCs (age and gender matched)	GFD for 1 y	Changes in depression score BA a GFD (subscore of PGWB)	1—unvalidated FU question (“strict diet” or “dietary lapses”)	
Vilppula et al. (2011) [62]	Finland Nationwide BA comparison	32 screen-detected biopsy-proven CD patients; age > 50 y	110 HCs recruited from neighbourhoods of CD patients (age and gender matched)	GFD for 1–2 y	Changes in depression score BA a GFD (subscore of PGWB)	2—with dietician 3—tTG, EmA 4—Vh/CrD (“Diet considered strict when there were no signs of dietary transgressions upon the interview. Occasional GFD defined as a gluten intake occurring less often than once in the month.”)	

NOTE: Studies in alphabetical order. Abbreviations: RCT, randomised controlled trial; BA, before-after; DB, double-blind; PC, placebo controlled; CO, cross-over; CD, coeliac disease; WA, wheat allergy; GFD, gluten-free diet; HC, healthy control; NC, no control group; IBS(-D), irritable bowel syndrome (diarrhoea-predominant); NCGS, non-coeliac gluten sensitivity; y, year; mo, month; wk, week; FU, follow-up; M-SDS, modified Zung Self-reported Depression Scale; HADS, Hospital Anxiety and Depression Scale; HDRS, Hamilton Depression Rating Scale; SCID-I, Structured Clinical Interview for DSM-IV Axis I Disorders; CCEI, Crown-Crisp Experiential Index; PGWB, Psychological General Well-Being Index; BDI, Beck Depression Inventory; STPI, State-Trait Personality Inventory; CDI, Children’s Depression Inventory; AGA, Anti-gliadin antibodies; EmA, anti-endomysial antibodies; tTG, Anti-tissue transglutaminase antibodies; Vh/CrD, villous height:crypt depth ratio; DGP, Deamidated Gliadin Peptide; NI, no information; NA, not applicable. * 1, Self-rated questionnaire; 2, Interview; 3, Serology; 4, Histology; 5, Food diary/record; 6, Other.

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
