# Peer review of "Mood Disorders and Gluten: It’s Not All in Your Mind! A Systematic Review with Meta-Analysis"

_nutrients, 2018, doi:10.3390/nu10111708_

Reviewer 1 Report

Busley et al wrote a meta-analysis aiming to investigate whether a gluten free diet (GFD) could be beneficial to improve mood disorders both in patients with CD or NCGS or in subjects without gluten related disorders. Main comments:

1)      A linguistic revision is necessary due to some typos.

2)      Page 1 line 14: I suggest to replace “all population” with “patients with or without gluten related disorders”.

3)      Page 1 lines 19-20 “with no difference...” this sentence is trunk. Between which groups?

4)      Page 1 lines 20-21: Authors are not allowed to conclude that symptoms worsened because p=0.22.

5)      Paragraph 3.4.1.2: a RD=-0.34 means a reduction of 34% of patients positive for depression. Please add this explanation in order to improve the readability of the paper.

6)      In figure 6 the cumulative analysis for all 3 studies is absent. Please add and comment.

7)      Paragraph 3.4.4: what do Authors mean for “GFD treated patients”? Patients with gluten related disorders? Indeed all subjects, even healthy controls, underwent a GFD.

8)      The analysis reported in fig 7B is the same of fig 4, so the same estimation of risk should be used (RD is preferred).

9)      The discussion is too long, therefore it may bore the reader. Please summarize it.

10)   Table 2 is a repetition of figures and should be better provided as supplementary material.

Author Response

Response to Reviewer 1 Comments

Many thanks for the feedback. Please see our responses and changes below.

Point 1: A linguistic revision is necessary due to some typos.

Response 1: Thank you for pointing this out; linguistics and typos have now been revised and hopefully corrected.

Point 2: Page 1 line 14: I suggest to replace “all population” with “patients with or without gluten related disorders”.

Response 2: Line 13: “All populations” has been replaced with “patients with or without gluten related disorders”

Point 3: Page 1 lines 19-20 “with no difference...” this sentence is trunk. Between which groups?

Response 3: We have endeavoured to make this clearer. Please see line 19: the sentence has been changed from “with no difference between GFD-treated patients and healthy controls” to “with no difference in mean scores between patients and healthy controls after 1 year”.

Point 4: Page 1 lines 20-21: Authors are not allowed to conclude that symptoms worsened because p=0.22.

Response 4: This is a good point – thank you. See lines 20-21 for the corrected sentence: “There was a tendency towards worsening symptoms for non-coeliac gluten sensitive patients during a blinded gluten challenge vs placebo”.

Point 5: Paragraph 3.4.1.2: a RD=-0.34 means a reduction of 34% of patients positive for depression. Please add this explanation in order to improve the readability of the paper.

Response 5: Line 318: we have changed the sentence to “there was a reduction of 31% of patients positive for depression after following a GFD” (reported results are slightly different as analyses have been changed to random-effects as per suggestion from another reviewer).

Point 6:  In figure 6 the cumulative analysis for all 3 studies is absent. Please add and comment.

Response 6: Thank you for your comment; we have detailed our explanation and reasoning below. Please see lines 343-345 which explains our choice for not including a cumulative analysis in figure 6; this explanation was already presented but has now been modified for clarity: “Two of the three studies used the same sample at different follow-up timepoints [52,58]. Moreover, as there is considerable variation in results and inconsistency in the direction of effect, no meta-analysis was conducted (Figure 6).”  This is following the suggestion from the Cochrane Handbook (version 5.1) section 5.9.3. which states: “If there is considerable variation in results, and particularly if there is inconsistency in the direction of effect, it may be misleading to quote an average value for the intervention effect.” We would be grateful if you could let us know whether you think a cumulative analysis for the two studies using different samples should be added regardless.

Point 7: Paragraph 3.4.4: what do Authors mean for “GFD treated patients”? Patients with gluten related disorders? Indeed all subjects, even healthy controls, underwent a GFD.

Response 7: We apologise if this was unclear. The healthy controls did not undergo a GFD in any of the included studies; the data for the controls are cross-sectional.

Point 8: The analysis reported in fig 7B is the same of fig 4, so the same estimation of risk should be used (RD is preferred).

Response 8: We agree; the analysis in Fig 7B (p. 21) has been changed to RD.

Point 9:  The discussion is too long, therefore it may bore the reader. Please summarize it.

Response 9: The discussion has now been edited down from around 2700 to 1700 words.

Point 10: Table 2 is a repetition of figures and should be better provided as supplementary material.

Response 10: Thank you for your suggestion. Nevertheless, Table 2 summarises the data from important comparisons as well as clearly grading the quality of the evidence across studies using the GRADE system; hence, we think it should remain in the main text.

Reviewer 2 Report

This is a well-written meta-analysis on an interesting topic and important topic. It has a lot of good information but is quite lengthy and needs to be condensed where possible (given the large amount of results presented.. Some points to consider:

Minor:

-you use a lot of acronyms which are difficult to keep track of (disease states to scales). Consider using less which would enhance the readability of your manuscript.

-I think you can say that you requested data from authors in the section, but I think it would be better to avoid stating in the results "data was requested from xxx but not answered"...instead just say you estimated the data from xxx(e.g., webplotdigitizer).

Introduction:

-consider altering sentence on line 31 to state it is a global health burden since there is approximately 300 million worldwide with depression (versus 425 million with diabetes which many would consider a global crisis).

-Since this article may be of interest to psychiatrists/psychologists, it may be useful to present a very brief background on gluten and gluten-related disorders

-The introduction has a lot of good information but it lacks a consistent, logical path likely because it is to long. It could be condensed without losing overall points you are making (for example, the paragraph on mechanistic evidence of gluten-related antibodies could be condensed to one or two sentences while giving appropriate references).

Methods:

-You should put the table of inclusion/exclusion criteria either in the body of the text or as a supplementary table. It was confusing going to the appendix for certain things, going to supplementary info, and so on

- Why did the authors choose a fixed effect analysis as the main method of analysis while reserving random-effects analysis as a sensitivity analysis? the effects of the intervention (diet) may not be consistent from individual-to-individual which perhaps is better suited for random effects

- DId you register your systematic review prior to performing it (PROSPERO, etc)?

-line 194 to 201, this is not entirely clear. Data was preferred from depression questionnaires that did not include questions on GI or eating habits while those that do (HAMD) were analyzed separately? You are treating it as missing data? please clarify and provide your reasoning (Line 443-444 has an explanation that you may consider putting forth earlier showing the potential bias of studies that include depression scales with GI questions).

Results:

-Line 226 is the first time that "observational" study is used. You continue to use it including in the tables but it isn't in the description of studies (section 3.2). This is a bit confusing since you excluded "cross-sectional" studies which many would think is a form of observational study. Consider defining what studies were observational in 3.2 or using a different term

-Line 248, eight studies were included but you just described 11 studies. Please make more clear the connection between the 11 studies and 8 studies included in the meta-analysis.

-similarly, in table 2 it says 7 studies for 953 subjects but the text says 8. Are you only stating the observational studies in this meta-analysis. It is hard to clearly understand what studies were included in each question of this table compared to the total studies included from the systematic review. This needs to be more clear.

-Consider including the p-values in the table as well

-Line 292, is the p-value correct? (p=.93 in text versus p=.85 in table)

-Line 367, several times in the results you exclude studies for some reason that is not defined in your exclusion criteria (other than inability to extract data). This gives the feeling that you are picking and choosing which data enters the meta-analysis rather than following your systematic procedure to collect pertinent. These reasons should be clearly defined in your methods.

-Line 372, consider re-wording "contrary to our findings", I think you a meaning figure 7a to 7b but this isn't clear.

-consider making some of your sensitivity analysis data available in your supplementary information

Discussion

-Line 433, why wasn't this study included in your study. Is it another meta-analysis?

-similar to your introduction, you discussion contains a lot of good information but needs to be shortened. Concentrate on the sections that you describe your overall findings compared to other meta-analyses or current state of knowledge rather than comparing the findings of your meta-analysis to various, individual studies.

Author Response

Response to Reviewer 2 Comments

Many thanks for the feedback. Please see our responses and changes below.

Point 1: You use a lot of acronyms which are difficult to keep track of (disease states to scales). Consider using less which would enhance the readability of your manuscript.

Response 1: We agree with this. Hence, the following acronyms have been removed: healthy controls (HC), gluten-related disorders (GRD), non-randomised studies (NRS), non-coeliac wheat sensitivity (NCWS), wheat allergy (WA), gluten-containing diet (GCD), risk of bias (RoB), major depressive disorder (MDD), gastrointestinal (GI), quality of life (QoL) and systematic review (SR). The more common acronyms for scales (BDI, etc), some disease states (IBS, CD and NCGS), and GFD remain.

Point 2: I think you can say that you requested data from authors in the section, but I think it would be better to avoid stating in the results "data was requested from xxx but not answered"...instead just say you estimated the data from xxx(e.g., webplotdigitizer).

Response 2: The following sentences have been removed: “The corresponding authors for each of these studies were contacted for the missing data but none replied” and “As data requests via email to the corresponding authors of Di Sabatino et al. were not answered…” (section 3.4.2.)

Introduction

Point 3: Consider altering sentence on line 31 to state it is a global health burden since there is approximately 300 million worldwide with depression (versus 425 million with diabetes which many would consider a global crisis).

Response 3: Thank you for this suggestion, we agree that it is an improved opening to the article. See lines 30-31; the first sentence now reads “Mood disorders are a global healthcare burden, with 300 million people now suffering from depression worldwide.”

Point 4: Since this article may be of interest to psychiatrists/psychologists, it may be useful to present a very brief background on gluten and gluten-related disorders.

Response 4: Having added these pieces of background we agree that the introduction flows a lot better and is generally clearer – thank you. See lines 36-53 for the added background to gluten and gluten-related disorders.

Point 5: The introduction has a lot of good information but it lacks a consistent, logical path likely because it is too long. It could be condensed without losing overall points you are making (for example, the paragraph on mechanistic evidence of gluten-related antibodies could be condensed to one or two sentences while giving appropriate references).

Response 5: We agree that the article was too long overall. From your suggestion, the introduction has now been rearranged and edited down from 1170 to 830 words.

Methods

Point 6: You should put the table of inclusion/exclusion criteria either in the body of the text or as a supplementary table. It was confusing going to the appendix for certain things, going to supplementary info, and so on.

Response 6: Thank you for this suggestion; we have actually now removed the table of inclusion/exclusion criteria (previously appendix A) as it didn’t add that much information and agree that it was slightly confusing; any extra information in the table has been moved to the body of the text (section 2.1, p. 3)

Point 7: Why did the authors choose a fixed effect analysis as the main method of analysis while reserving random-effects analysis as a sensitivity analysis? the effects of the intervention (diet) may not be consistent from individual-to-individual which perhaps is better suited for random effects

Response 7: On hindsight we agree with your comment; all analyses have now been changed to random-effects.

Point 8: Did you register your systematic review prior to performing it (PROSPERO, etc)?

Response 8: Unfortunately we did not register our review in time before data collection took place and hence it was not accepted.

Point 9: line 194 to 201, this is not entirely clear. Data was preferred from depression questionnaires that did not include questions on GI or eating habits while those that do (HAMD) were analyzed separately? You are treating it as missing data? please clarify and provide your reasoning (Line 443-444 has an explanation that you may consider putting forth earlier showing the potential bias of studies that include depression scales with GI questions).

Response 9: Thank you for pointing out that this is unclear. Please see lines 152-155 for the following explanation (now moved to section 2.4. Statistical Analysis rather than section 2.5. Missing Data): ‘Where possible, only depressive outcome data from questionnaires not biased towards physical illness, with no questions related to gastrointestinal health and eating habits, were synthesized in meta-analysis; data from studies using biased questionnaires containing these types of questions were reported separately.’ See lines 216-226 for information regarding which questionnaire was used by each study (now detailed under 3.2. Study Characteristics).

Results

Point 10: Line 226 is the first time that "observational" study is used. You continue to use it including in the tables but it isn't in the description of studies (section 3.2). This is a bit confusing since you excluded "cross-sectional" studies which many would think is a form of observational study. Consider defining what studies were observational in 3.2 or using a different term

Response 10: We agree that this terminology is confusing; ‘observational’ has now been changed to either ‘longitudinal’ or ‘non-randomised’ throughout the paper.

Point 11: Line 248, eight studies were included but you just described 11 studies. Please make more clear the connection between the 11 studies and 8 studies included in the meta-analysis.

Response 11: To aid clarity, we have added the following explanation in lines 290-292: ‘two studies were not eligible for meta-analysis [59,66] and one study [52] was a follow-up of the same sample from another study [58]

Point 12: Similarly, in table 2 it says 7 studies for 953 subjects but the text says 8. Are you only stating the observational studies in this meta-analysis. It is hard to clearly understand what studies were included in each question of this table compared to the total studies included from the systematic review. This needs to be more clear.

Response: Thank you for pointing this out as it is an error; Table 2 now correctly states ‘1 RCT and 7 non-randomised studies’ rather than the previous ‘7 observational studies’

Point 13: Consider including the p-values in the table as well

Response 13: Good point; p-values have now been added to Table 2.

Point 14: Line 292, is the p-value correct? (p=.93 in text versus p=.85 in table)

Response 14: This was another error on our behalf; in any case, these have been corrected to the values from random-effects analyses.

Point 15: Line 367, several times in the results you exclude studies for some reason that is not defined in your exclusion criteria (other than inability to extract data). This gives the feeling that you are picking and choosing which data enters the meta-analysis rather than following your systematic procedure to collect pertinent. These reasons should be clearly defined in your methods.

Response 15: We are grateful for this suggestion and agree that the exclusion criteria needed further definition; see section 2.1 for the further criteria for inclusion/exclusion to meta-analysis that we have now added.

Point 16: Line 372, consider re-wording "contrary to our findings", I think you a meaning figure 7a to 7b but this isn't clear.

Response 16: Now line 378; this sentence has been reworded.

Point 17: Consider making some of your sensitivity analysis data available in your supplementary information

Response 17: We agree that the sensitivity analysis data should be made available and have done now in supplementary file 6.

Discussion

Point 18: Line 433, why wasn't this study included in your study. Is it another meta-analysis?

Response 18: Thank you for pointing this out as this was an error; the cited study actually measured depression at baseline rather than after a GFD. The reason for study exclusion is given in Table S8, supplementary file 3. In any case, this part of the discussion has been removed due to editing down (please see our response to your next point).

Point 19: Similar to your introduction, you discussion contains a lot of good information but needs to be shortened. Concentrate on the sections that you describe your overall findings compared to other meta-analyses or current state of knowledge rather than comparing the findings of your meta-analysis to various, individual studies.

Response 19: The discussion has been edited down from approx. 2700 to 1700 words

Reviewer 3 Report

Thank you for such an extensive and thorough examination of this important topic. I believe it will be of great interest to the broader healthcare community. It would significantly benefit from being shortened and 'wordsmithed' so some of the important messages and the 'flow' of the paper are clearer and more concise. 

Firstly, I would suggest you provide clearer definitions of the GRDs that have a medical indication for  a GFD, and a defintion of NCGS. A brief sentence outlining what a GFD and FODMAP diet involves would help those less familiar with the topic . I struggled to identifiy how you organised your sub-groups of interest into CD, NCGS and IBS until much later in the paper. I was also interested as to why you didnt include studies about other extraintestinal conditions known to be related to gluten including dermatitis herpetiformis and gluten ataxia.  

Some of the valuable conclusion points would be better placed in the dicussion and the conclusion itself needs to align more with the aims of the review and what you have stated in your abstract conclusion.

I wish you all the best with further research you may being conducting on this topic and look forward to seeing more results in due course. From reading your paper, a study that includes patients with mild to moderate depression, who have GRDs excluded at baseline and follow a 12 month GFD would be a very interesting study.

Author Response

Response to Reviewer 3 Comments

Many thanks for the feedback and comments. Please see our responses and changes below.

Point 1: It would significantly benefit from being shortened and 'wordsmithed' so some of the important messages and the 'flow' of the paper are clearer and more concise. 

Response 1: We agree that article was too long; the introduction and discussion have now been edited down, with around 1500 words cut out overall.

Point 2: Firstly, I would suggest you provide clearer definitions of the GRDs that have a medical indication for a GFD, and a definition of NCGS.

Response 2: Thank you for this comment and the next – having added these points we think the introduction now flows a lot better. See lines 36-53: a background to gluten and gluten-related disorders have been added.

Point 3: A brief sentence outlining what a GFD and FODMAP diet involves would help those less familiar with the topic.

Response 3: GFD, lines 40-41; low-FODMAP, lines 70-72.

Point 4: I struggled to identify how you organised your sub-groups of interest into CD, NCGS and IBS until much later in the paper.

Response 4: Thank you for pointing out that this is unclear; we didn’t organise our subgroups based on the type of diagnosed disorder, although we did conduct a sub-analysis comparing CD patients to non-CD patients. It just so happened that the two short-term double-blind RCTs were on NCGS populations and were hence analysed as a separate comparison due to the short-term nature of the invention.

Point 5: I was also interested as to why you didn’t include studies about other extraintestinal conditions known to be related to gluten including dermatitis herpetiformis and gluten ataxia.  

Response 5: We agree that this needed to be further clarified; lines 427-430 have been added to explain that no studies were excluded based on conditions or disorders – gluten-related or otherwise: ‘Although diagnosed conditions or disorders were not an exclusion criteria for our study, only studies on populations with CD, IBS or NCGS were identified through our searches. No other gluten-related disorder, such as dermatitis herpetiformis or gluten ataxia, nor any other condition, such as major depressive disorder, were identified.’

Point 6: Some of the valuable conclusion points would be better placed in the discussion and the conclusion itself needs to align more with the aims of the review and what you have stated in your abstract conclusion.

Response 6: This was very helpful; in line with your suggestions, we have now more clearly outlined our aims and objectives in the last paragraph of our introduction, and the conclusion is now more aligned with the aims of the review and the abstract conclusion. The discussion has been made more concise and has been reorganised based on some of the points in the conclusion.

Round  2

Reviewer 1 Report

The paper may be accepted

Reviewer 2 Report

I am satisfied with the reviewer revisions. They have done a thorough job addressing reviewer comments